# Comparative Analysis of Methicillin-Resistant *Staphylococcus pseudintermedius* Prevalence and Resistance Patterns in Canine and Feline Clinical Samples: Insights from a Three-Year Study in Germany

**DOI:** 10.3390/antibiotics13070660

**Published:** 2024-07-17

**Authors:** Leonie Feuer, Stefanie Katharina Frenzer, Roswitha Merle, Wolfgang Bäumer, Antina Lübke-Becker, Babette Klein, Alexander Bartel

**Affiliations:** 1Institute of Pharmacology and Toxicology, School of Veterinary Medicine, Freie Universität Berlin, Koserstraße 20, 14195 Berlin, Germany; leonie.feuer@fu-berlin.de (L.F.); wolfgang.baeumer@fu-berlin.de (W.B.); 2Institute of Veterinary Epidemiology and Biostatistics, School of Veterinary Medicine, Freie Universität Berlin, Königsweg 67, 14163 Berlin, Germany; katharina.frenzer@fu-berlin.de (S.K.F.); roswitha.merle@fu-berlin.de (R.M.); alexander.bartel@fu-berlin.de (A.B.); 3Veterinary Centre for Resistance Research (TZR), Freie Universität Berlin, Robert-von-Ostertag-Straße 8, 14163 Berlin, Germany; 4Institute of Microbiology and Epizootics, School of Veterinary Medicine, Freie Universität Berlin, Robert-von-Ostertag-Straße 7, 14163 Berlin, Germany; 5LABOKLIN GmbH und Co. KG, Steubenstraße 4, 97688 Bad Kissingen, Germany; b.klein@laboklin.com

**Keywords:** MRSP, antimicrobial resistance, AMR monitoring, companion animals, retrospective

## Abstract

The emergence of methicillin-resistant *Staphylococcus pseudintermedius* (MRSP) presents a significant public health concern globally, particularly within veterinary medicine. MRSP’s resistance to multiple antibiotics is limiting treatment options and potentially leading to severe infections in companion animals. This study aimed to understand antimicrobial resistance in dogs and cats, focusing on MRSP resistance patterns and its prevalence in Germany. We analyzed results of bacterial diagnostic samples from canines and felines, sourced from a German veterinary diagnostic microbiology laboratory between 2019 and 2021. This dataset included samples from 3491 veterinary practices, covering 33.1% of veterinary practices and clinics in Germany. MRSP rates were detailed by host species, sample types and co-resistance patterns. Analysis of 175,171 bacterial examination results revealed *S. pseudintermedius* in 44,880 samples, yielding a 25.6% isolation rate. *S. pseudintermedius* was more prevalent in dogs (35.0%) than cats (3.6%). Methicillin resistance was found in 7.5% of all *S. pseudintermedius* isolates. MRSP prevalence was higher in feline samples (16.1%, 95% CI 14.4–17.8) compared to canine samples (7.1%, 95% CI 6.8–7.0). *S. pseudintermedius* showed high resistance rates to ampicillin (cats: 48.6%, dogs: 67.6%) and clindamycin (cats: 37.2%, dogs: 32.7%), while MRSP exhibited high co-resistance to clindamycin (cats: 82.8%, dogs: 85.4%) and sulfamethoxazole + trimethoprim (cats: 66.4%, dogs: 66.2%). Our study revealed distinct resistance patterns of MRSP in cats compared to dogs, highlighting the need for tailored treatment approaches and the importance of antimicrobial resistance surveillance.

## 1. Introduction

In 2021, the European Food Safety Authority (EFSA) identified *Staphylococcus (S.) pseudintermedius*, as one of the three most relevant antimicrobial resistant bacteria in the EU that constitute a threat to the health of dogs and cats [1]. *S. pseudintermedius*, commonly found as a commensal on the skin and mucous membranes of dogs and cats, is the predominant bacterial pathogen found in clinical canine samples, often associated with various infections such as those affecting the skin, ear, and post-surgical sites [2,3,4].

The emergence of antimicrobial resistance (AMR), driven by natural evolutionary responses to antimicrobial exposure [5,6], has led to the widespread prevalence of methicillin-resistant strains of *S. pseudintermedius* (MRSP) among both healthy and diseased dogs and cats [3,4]. Beyond methicillin, MRSP strains commonly exhibit co-resistance to multiple classes of antimicrobials, including fluoroquinolones, macrolides, tetracyclines, and aminoglycosides [3,7,8].

The global emergence of MRSP has significantly complicated treatment strategies, giving rise to challenging infections in companion animals, such as superficial pyoderma, otitis externa, urinary tract infections, and more severe systemic infections including septicemia and endocarditis. This presents a substantial clinical challenge in veterinary medicine [3,4,8,9,10].

To ensure effective treatment with antibiotics, a range of strategies aimed at mitigating AMR while promoting prudent use of antibiotics operates both at national and international levels. In Germany, at the national level, the Veterinary House Dispensary Ordinance (TÄHAV) is in place. With its amendment in 2018, the TÄHAV aims to combat antibiotic resistance by limiting antibiotic treatments to necessary cases, focusing on preserving the effectiveness of third- and fourth-generation cephalosporins and fluoroquinolones. It therefore mandates the availability or pending status of bacterial culture and susceptibility results when prescribing these antibiotics, underscoring the importance of informed antibiotic selection [11]. Additionally, EU legislation imposes restrictions on certain antimicrobials in veterinary medicine [12]. Beyond regulatory frameworks, strategies encompass guidelines for antimicrobial usage and the implementation of antimicrobial stewardship programs aimed at optimizing antibiotic prescribing practices [7,13,14].

Accurate data on both the prevalence and time trends of AMR are essential for the formulation of guidelines and legislation concerning antibiotic usage. Presently, monitoring of AMR in bacterial species in animals is not organized at a European level. However, initiatives such as the European Antimicrobial Resistance Surveillance network in Veterinary medicine (EARS-Vet) have been launched to strengthen the European One Health AMR surveillance approach [15].

In Germany, the national veterinary resistance monitoring system, GERM-Vet, is in place. It systematically examines isolates from diverse animal species obtained from various laboratories according to a sampling plan, but voluntary participation keeps the number of isolates manageable. The limited sample sizes impede data evaluation for certain species and indications. In 2021, the dataset included a total of 199 isolates from cats and 450 isolates from dogs, with 147 of these isolates identified as canine *S. pseudintermedius* [16]. While these analyses provide valuable insights including whole-genome sequencing data, the representativeness of prevalence data remains challenging.

Broad-scale MRSP prevalence figures in Germany can be found in a 2023 study by Loeffler et al., which evaluated MRSP prevalence among clinical samples from dogs, tracking trends following the enactment of antimicrobial prescribing legislation in Germany [17]. While their study focused exclusively on *S. pseudintermedius* from dogs, research indicates that *S. pseudintermedius*, particularly MRSP, is also a concern in cats [18,19].

To our knowledge, ours is the first study in Germany to present a substantial dataset on MRSP in diseased cats alongside dogs, enabling a comparative analysis of MRSP prevalence, sample types, and prevailing co-resistance patterns from 2019 to 2021 between these two species. The study aimed to establish a baseline for MRSP prevalence in dogs and cats, support evidence-based decision-making in empirical therapy, and enable the interpretation of future trends.

## 2. Results

Overall, our analysis encompassed the results of 175,171 bacteriological examinations of both feline and canine specimens. Out of these, 27,917 samples (19,154 from canines and 8763 from felines) showed no growth of specific pathogenic bacterial species. The distribution of sample types was as follows: 67,293 skin/soft tissue samples, 16,111 wound samples, 21,398 respiratory tract samples, 20,907 urogenital tract samples, and 49,462 other samples.

*S. pseudintermedius* was identified in 44,880 samples, constituting 25.6% of the total examined. The occurrence rates varied notably between cats (3.6%) and dogs (35.0%). *S. pseudintermedius* was isolated in the following proportions across different sample types: 33.2% from skin/soft tissue samples (dogs: 41.3%, cats: 4.6%), 22.2% from wound samples (dogs: 32.5%, cats: 6.3%), 15.3% from respiratory tract samples (dogs: 29.8%, cats: 2.4%), 15.3% from urogenital tract samples (dogs: 44.8%, cats: 1.9%), and 25.3% from other samples (dogs: 27.6%, cats: 3.5%). Detailed sample counts for *S. pseudintermedius* isolation categorized by specific organ systems are presented in Table 1.

In total, 7.5% (95% CI 7.2–7.7, n = 3346) of the investigated *S. pseudintermedius* strains exhibited phenotypic oxacillin resistance. Notably, *S. pseudintermedius* isolated from cats demonstrated a prevalence of 16.1% (95% CI 14.4–17.8, n = 300), surpassing the prevalence of 7.1% (95% CI 6.8–7.0, n = 3046) observed in samples from dogs. Our analysis revealed a stable MRSP rate across the years 2019 to 2021, as illustrated in Figure 1.

MRSP resistance rates in dogs exhibited minimal variation across sample types, except for wound samples, which showed the highest methicillin-resistance rate among canine *S. pseudintermedius* samples, at 15.2%. Canine samples from the respiratory tract, skin and soft tissue, urogenital tract, and other showed methicillin-resistance below 7.5%. In cats, *S. pseudintermedius* displayed elevated resistance rates, particularly in samples from wounds (24.7%) and urogenital tract infections (21.4%). In contrast, feline samples from the respiratory tract and skin/soft tissue exhibited comparatively lower resistance rates (12.6% and 11.0%, respectively). There were no major differences in the MRSP prevalence within an organ system between 2019 and 2021 (refer to Figure 2).

Within all *S. pseudintermedius* samples, canine samples exhibited significantly higher resistance rates to ampicillin at 67.6% (95% CI 67.2–68.1) compared to cats at 48.6% (95% CI 46.1–51.0). Canine samples also exhibited significantly higher resistance rates to chloramphenicol at 6.9% (95% CI 6.6–7.2), compared to cats at 5.3% (95% CI 4.3–6.5). Conversely, feline samples demonstrated significantly greater resistance to several other antimicrobials. Resistance to clindamycin was observed in 37.2% (95% CI 35.0–39.4) of feline samples, as opposed to 32.7% (95% CI 32.2–33.1) in canine samples. Similarly, resistance rates were higher in feline samples for cephalexin (16.8% vs. 7.9%), enrofloxacin (11.4% vs. 5.7%), and sulfamethoxazole + trimethoprim (15.4% vs. 9.8%). Both canine and feline isolates exhibited low resistance rates to doxycycline, rifampicin, and gentamicin, with less than 3.8% for all three antimicrobials. (Figure 3).

MRSP isolates generally exhibited resistance to all tested beta-lactam antibiotics. Co-resistance to non-beta-lactam antibiotics was notably high for clindamycin (85.2%, 95% CI 83.9–86.4), sulfamethoxazole + trimethoprim (66.3%, 95% CI 64.6–67.9), and enrofloxacin (50.5%, 95% CI 48.8–52.2). The underlying data are available as Appendix A.

## 3. Discussion

This study extends our prior research on AMR data in companion animals in Germany [22] and aims to shed light on the prevalence and resistance patterns of MRSP in clinically diseased dogs and cats between 2019 and 2021. Our analysis encompasses a significant dataset consisting of 175,171 samples received from approximately one-third (33.1%) of all registered German veterinary practices and clinics (10,558) as of 2021 [23]. With a total of 44,880 *S. pseudintermedius* isolates, including 1864 from cats, our study offers a comprehensive analysis contrasting MRSP in canine and feline samples in Germany from 2019 to 2021.

In our study, we found several notable differences between cats and dogs in the prevalence of MRSP across different sample types and resistance patterns. As noted by Mendandro et al., comparing different studies is challenging due to the influence of geographic origin, sample type, and the population studied on MRSP occurrence. Variations in prevalence data can also result from differences in study design, identification methods, sampling techniques, animal health status, and other factors [24]. Notably, the literature indicates that *S. pseudintermedius* is commonly found in both dogs and cats, although it is more frequently isolated in dogs [18,25,26], as supported by our study’s findings, where lower case numbers for feline *S. pseudintermedius* isolates (3.6% of 52,340 feline samples) were observed. This observation aligns with previous studies, which further suggested a higher prevalence of *S. pseudintermedius* colonization in cats housed with dogs, underscoring that it should not be evaluated as natural feline microbiota [19]. However, our study also shows that when *S. pseudintermedius* is present in cats, there is a high risk for methicillin resistance. It is, therefore, important to also collect comprehensive MRSP prevalence data from cats, allowing evaluation of resistance patterns and trends.

### 3.1. MRSP Prevalence in Cats and Dogs

The feline MRSP prevalence in our study from 2019 to 2021 was 16.1%. With this, it is higher than the 5% reported in sick cats during the same period in Poland [25]. However, it is significantly lower than the 43.8% (*S. pseudintermedius* n = 6828) prevalence reported by Sobkowich et al. for the USA during the same period [18]. In northern Italy, the feline MRSP prevalence was 20% (*S. pseudintermedius* n = 5) in the study period 2015–2016 [24]. Previous studies in Germany showed feline MRSP rates of 6.6% (*S. pseudintermedius* n = 91) in 2007 [27], 94.1% (*S. pseudintermedius* n = 17) in the study period 2010–2011 [28], and 27.3% (*S. pseudintermedius* n = 11) in the study period 2017–2018 [29]. These discrepancies highlight the challenges of comparing feline MRSP rates due to generally smaller sample sizes in these previous European studies.

The canine MRSP prevalence in our study was 7.1%. This prevalence is higher than the 2.5% reported in Poland [25], but significantly lower than the 32.4% reported in the USA [18] and the 32.7% in northern Italy [24]. Compared to previous studies in Germany, our canine prevalence closely matches the 6.8% reported by Ruscher et al. [27] and the 8–12% range reported by the GERM-Vet surveillance system [16,30]. Loeffler et al. also reported a prevalence of 7.1% [17], which is expected given the overlap in sample sets with our study. These similarities suggest that our findings are consistent with previous data within Germany. The relatively higher sample sizes in canine studies likely contribute to the more representative findings, in contrast to the often smaller feline sample sizes. Variations in MRSP prevalence between countries could be influenced by differences in antibiotic use practices or transmission rates in veterinary practices and clinics. However, the specific causes of these differences cannot be explained by this study.

Notable is the large difference in MRSP prevalence between dogs and cats. *S. pseudintermedius* seems to be uniquely adapted to dogs, which is a common member of the canine flora and a common pathogen for skin, urinary tract, and other infections in dogs [9]. We can only speculate that this adaption changes the pathomechanism of *S. pseudintermedius* infections in dogs, indirectly resulting in lower resistance rates. This difference, however, does not seem to apply to wound samples (see Section 3.2 below). Especially for pyoderma, one can assume a difference in pathomechanism between dogs and cats, with canine pyoderma being one of the most common reasons for presentation in a veterinary practice [9]. Other factors, including differences in the antibiotics used for each species and the tendency of cats to visit the vet only when they are seriously ill, as they often hide their symptoms, can also have an effect on the resistance rate. As there are hardly any studies comparing the MRSP rates of cats and dogs, we can only discuss possible influencing factors to a limited extent. Further research is needed to determine whether MRSP prevalence is consistently higher in cats and to explore the underlying reasons.

### 3.2. Feline and Canine Sample Types

In cats, the highest MRSP prevalence was found in wound samples and urinary tract samples, both exceeding 20%, and in respiratory tract samples, which exceeded 12%. While antibiotics for urinary tract infections and respiratory tract infections are administered systemically, skin and soft tissue infections can be treated locally. Higher resistance rates may indicate that previous antibiotic treatments were not fully effective, possibly due to inadequate administration, which is particularly challenging with oral medications in cats. This can promote the development of resistance [5].

Within canine *S. pseudintermedius* samples, wound samples exhibited particularly high MRSP prevalence. This finding aligns with previous studies [27,29,31], which indicate that *S. pseudintermedius* can cause post-operative wound infections in dogs. Windahl et al. identified *S. pseudintermedius* as the most prevalent pathogen among canine surgical wound infections in Sweden [32]. Furthermore Viegas et al. reported that MRSP was more likely to be isolated from surgical site infections than from pyoderma or other site infections [33]. These findings are consistent with our study, where surgical site infections were categorized under the sample type wound.

### 3.3. Resistance Patterns in Feline and Canine S. pseudintermedius

While MRSP in dogs and cats exhibit similar resistance patterns, notable differences emerge when considering all *S. pseudintermedius.* In feline *S. pseudintermedius* samples, resistance to ampicillin and clindamycin showed the highest values of 48.6% and 37.2%, respectively, indicating that a significant proportion of *S. pseudintermedius* infections are resistant to commonly prescribed first-line antimicrobials [34]. *S. pseudintermedius* in cats furthermore exhibited greater resistance to cephalexin (16.8%), enrofloxacin (11.4%), and sulfamethoxazole + trimethoprim (15.4%) compared to dogs. These variations align with the findings of Feßler et al. [29]. In a Polish study, clinically ill cats demonstrated even higher resistance rates: 84.6% for ampicillin, 46.2% for enrofloxacin, and 38.5% for sulfamethoxazole + trimethoprim [19].

Canine *S. pseudintermedius* samples in our study showed even higher resistance rates to ampicillin at 67.6% compared to cats. Conversely, canine samples showed up to 8% less resistance to cephalexin, enrofloxacin, and sulfamethoxazole + trimethoprim compared to cats in our study. When comparing canine samples to data from GERM-Vet, our study shows slightly lower resistance levels. In GERM-Vet, resistance rates for canine skin and soft tissue samples in 2020 and 2021 were 73% and 69%, respectively, for ampicillin, 11% and 10%, respectively, for enrofloxacin, and 11% and 13%, respectively, for sulfamethoxazole + trimethoprim [16,30]. In our study, MRSP in both canine and feline samples exhibited high co-resistance to almost all tested antimicrobials. These findings are consistent with previous research, which furthermore demonstrated that samples with prior antimicrobial treatment had increased co-resistance [16,33]. Changed resistance patterns may be a response to the increased use of antibiotics that were rarely used in the past but are now being utilized due to emerging resistances [26]. Consequently, ongoing monitoring and mapping of resistance patterns in dogs and cats are of significant interest to understand these trends better.

### 3.4. AMR Monitoring

The monitoring of AMR is crucial in both human and veterinary medicine. Pathogens like MRSP, although primarily found in dogs and cats, pose significant risks to humans [35]. The mec cassettes in MRSP can be transferred to other staphylococci, potentially those more adapted to humans, through horizontal gene exchange [36]. To expand the scope of national AMR monitoring, laboratory data such as the dataset used in this study could prove to be a good source, as also described in our primary research on MRSA [22] and third generation resistant *Escherichia coli*.

Research indicates that regulatory measures can significantly impact AMR rates [17,37,38]. However, these measures must be evidence-based and should not compromise animal health. Such legislation affects the veterinary profession’s autonomy, sometimes leading to opposition [39]. Efforts are essential to provide education on pathogen-specific resistance rates alongside the development of comprehensive therapy guidelines. These measures are crucial to empower veterinarians with the necessary knowledge for making informed decisions, particularly in the management of bacterial infections involving resistant pathogens such as MRSP in cats and dogs.

### 3.5. Limitations

One limitation of this study pertains to the lacking history regarding antibiotic pre-treatment in the laboratory record, which could provide insights into the varied resistance patterns between isolates from pre-treated and untreated animals. Typically, when initial antibiotic therapy fails or when a particular antibiotic necessitating an antibiogram is considered [11], swabs are sent to the laboratory for analysis [22,40]. Consequently, the isolates being evaluated in this study probably exhibit greater resistance compared to pathogens causing infections already responsive to first-line antibiotic therapy [16,18]. But, even if the resistance observed in the laboratory diagnostics settings are likely to be greater than in the general population, these findings indicate that MRSP is a present issue in companion animals in Germany.

In our study, data regarding repeated or duplicated isolates were unavailable, potentially leading to inflated observed resistance rates. However, our data provider, based on expert opinion, assessed the duplication rate to be low, accounting for less than 1% of isolates, primarily due to the collection of data predominantly from outpatient visits.

In conclusion, our study provides comprehensive data on the prevalence of MRSP in clinically diseased cats and dogs across Germany. While the MRSP prevalence in dogs aligns with previous studies, our dataset on *S. pseudintermedius* in cats reveals higher prevalence and distinct resistance patterns compared to dogs. This underscores the need for tailored treatment approaches and highlights the importance of such AMR surveillance.

## 4. Materials and Methods

### 4.1. Samples

Samples originated from clinically diseased cats and dogs across Germany and were analyzed by Laboklin (Bad Kissingen, Germany), one of the accredited specialist laboratories for veterinary diagnostics in the country. During the timeframe from January 2019 to December 2021, 3491 veterinary practices and clinics submitted canine and feline samples to Laboklin. The dataset provided by the laboratory included information on the species, sample types, identified pathogens, and antimicrobial susceptibility testing. We retrospectively analyzed all results of bacterial diagnostic samples from this period.

### 4.2. Bacterial Identification and Antimicrobial Susceptibility Testing (AST) of S. pseudintermedius

Bacterial species were identified by the laboratory based on culture morphology, hemolysis, and MALDI-TOF-MS (Bruker Daltonics, Bremen, Germany) following established protocols. Antimicrobial susceptibilities were determined via broth microdilution testing around breakpoints according to Clinical and Laboratory Standards Institute (CLSI) Performance Standards documents, utilizing the MICRONAUT System Merlin (MERLIN GmbH, Bornheim-Hersel, Germany). This involved automated photometric evaluation of customized microtiter plates to ascertain the minimum inhibitory concentration (MIC). Within the scope of this study, the evaluation of feline and canine *S. pseudintermedius* isolates focused on several antimicrobial substances, including oxacillin, ampicillin, cephalexin, chloramphenicol, clindamycin, enrofloxacin, gentamicin, doxycycline, rifampicin, and sulfamethoxazole + trimethoprim. MRSP was identified based on oxacillin resistance, with chromagar used as secondary method for unclear results.

### 4.3. AST Classification

MIC interpretation (S—I—R) was conducted in accordance with standardized procedures outlined by the Clinical and Laboratory Standards Institute (CLSI), referencing CLSI documents Vet01S-Ed6 and M100Ed33 [20,21]. MRSP identification was based on an oxacillin MIC ≥ 0.5 μg/mL as specified by Vet01S-Ed6. If oxacillin resistance was detected in *S. pseudintermedius*, isolates were deemed resistant to all beta-lactam antibiotics, beta-lactam combination agents, cephems, and carbapenems, as outlined in the CLSI document Vet01S-Ed6 [20]. For all canine isolates, dog-specific breakpoints as outlined in the CLSI document Vet01S-Ed6 were applied whenever available. In cases where dog-specific breakpoints for urinary tract infection (UTI) were not provided, we adopted those designated for cat UTI. If no breakpoint for cat UTI was available, we used the breakpoint provided for dog skin and soft tissue (SST) infections. For all feline isolates, we employed cat-specific breakpoints from document Vet01S-Ed6 whenever available. In cases where cat UTI breakpoints were missing, we assigned the breakpoints designated for dog UTI. If neither cat UTI nor cat SST breakpoints were available, we utilized breakpoints for dog SST infections. In cases where species-specific breakpoints were lacking, we resorted to human-specific clinical breakpoints outlined in the CLSI document M100. These were applied for gentamicin, doxycycline, chloramphenicol, rifampicin, and sulfamethoxazole + trimethoprim.

### 4.4. Data Processing and Statistical Analysis

The samle types from the dataset were classified into five groups based on anatomical origin.

Skin and soft tissue (e.g., ear swabs)Wounds (e.g., swabs from surgery wounds and abscesses)Respiratory tract (e.g., nasal swabs and bronchoalveolar lavage)Urogenital tract (e.g., urinary samples and vaginal swabs)Other (e.g., unknown and gastrointestinal swabs)

All statistical analyses were conducted using R version 4.2.2 (R Foundation Vienna). Isolates and antimicrobial substances were identified using the R package AMR [41]. AST classification was performed in R with previously mentioned breakpoints. Results are presented with 95% Wilson confidence intervals (95% CI). Non-overlapping confidence intervals can be considered significantly different.

## Figures and Tables

**Figure 1 antibiotics-13-00660-f001:**
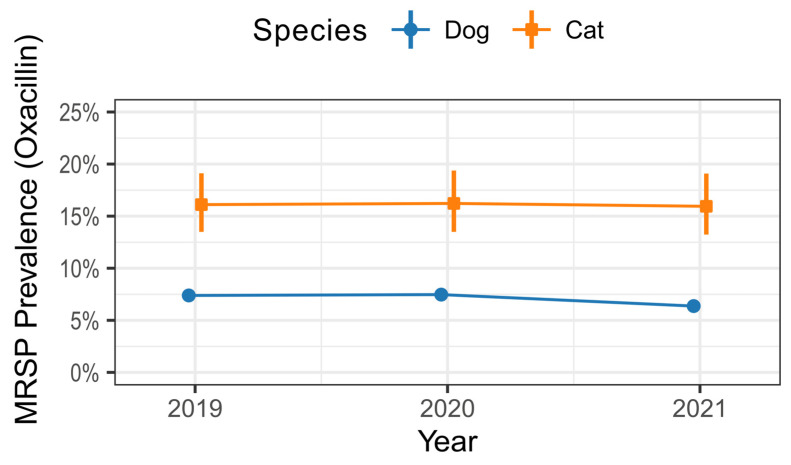
Proportion (%) of MRSP (defined by oxacillin resistance) among *S. pseudintermedius* infections (n = 44,880) in dogs and cats per year. Error bars represent 95% confidence intervals.

**Figure 2 antibiotics-13-00660-f002:**
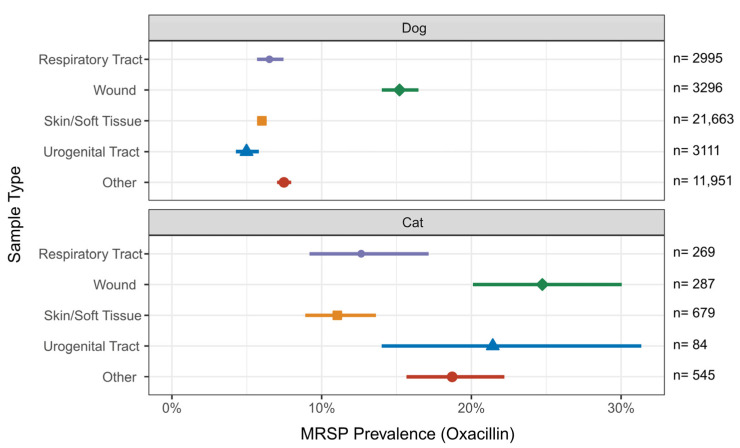
MRSP prevalence (defined by oxacillin resistance) in % among *S. pseudintermedius* infections of 5 different sample origins (respiratory tract, wound, skin and soft tissue, urogenital tract, and other) of dogs and cats per year. Error bars represent 95% confidence intervals.

**Figure 3 antibiotics-13-00660-f003:**
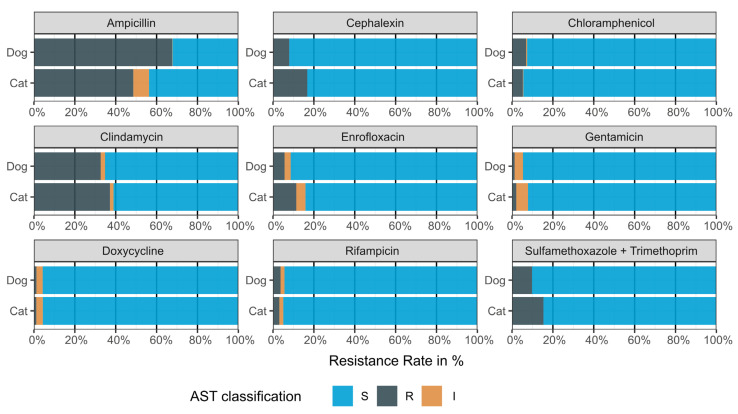
Resistance to various antimicrobial substances within all canine and feline *S. pseudintermedius* samples (n = 44,880) compared between species. MIC interpretation (S—I—R) was performed following the CLSI guidelines referencing documents Vet01S-Ed6 and M100Ed33 [20,21]. The breakpoints used were the following: Ampicillin (canine and feline urinary tract infections (UTI) and feline non-UTI): S ≤ 0.25, R ≥ 1, (canine non-UTI): S ≤ 0.25, R ≥ 0.5; cephalexin (canine and feline UTI and non-UTI): S ≤ 2, R ≥ 4; chloramphenicol (canine and feline UTI and non-UTI): S ≤ 8, R ≥ 32; clindamycin (canine and feline UTI and non-UTI): S ≤ 0.5, R ≥ 4; enrofloxacin (canine and feline UTI and non-UTI): S ≤ 0.5, R ≥ 4; gentamicin (canine and feline UTI and non-UTI): S ≤ 4, R ≥ 16; doxycycline (canine and feline UTI and non-UTI): S ≤ 4, R ≥ 16; rifampicin (canine and feline UTI and non-UTI): S ≤ 1, R ≥ 4; sulfamethoxazole + trimethoprim (canine and feline UTI and non-UTI): S ≤ 2/38, R ≥ 4/76. Abbreviations: AST: antimicrobial susceptibility testing, S: susceptible, I: intermediate, R: resistant.

**Table 1 antibiotics-13-00660-t001:** Total number and percentages of canine and feline samples per host species and year; total number and percentages of *S. pseudintermedius* overall and with regard to the assigned organ system per host species and year.

	Overall	Dog	Cat
Samples	175,171	122,831	52,340
Samples with *S. pseudintermedius* isolated	44,880	43,016	1864
Year (%)	44,880 (100)		
2019	16,145 (36.0)	15,487 (36.0)	658 (35.3)
2020	13,944 (31.1)	13,340 (31.0)	604 (32.4)
2021	14,791 (32.9)	14,189 (33.0)	602 (32.3)
Sample type (%)	44,880 (100)		
Skin/soft tissue	22,342 (49.8)	21,663 (50.4)	679 (36.4)
Wound	3583 (8.0)	3296 (7.7)	287 (15.4)
Respiratory tract	3264 (7.3)	2995 (7.0)	269 (14.4)
Urogenital tract	3195 (7.1)	3111 (7.2)	84 (4.5)
Other	12,496 (27.8)	11,951 (27.7)	545 (29.3)

## Data Availability

Data generated or analyzed during this study are included in this published article and its Appendix A. The file contains information on species, year, sample type, and AST classifications. Geographical information is not made available for reasons of confidentiality.

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
