# Peer review of "Comparative Analysis of Methicillin-Resistant Staphylococcus pseudintermedius Prevalence and Resistance Patterns in Canine and Feline Clinical Samples: Insights from a Three-Year Study in Germany"

_antibiotics, 2024, doi:10.3390/antibiotics13070660_

Round 1

Reviewer 1 Report

Comments and Suggestions for Authors

The authors compared analysis of MRSP prevalence and AMR patterns in dogs and cats in Germany: a retrospective study between 2019-2021. 

The manuscript is written well and easy to understand it. The authors show the impact of the total number of S. pseudintermedius in 44880 samples in Germany and their AMR patterns which are useful data for both veterinarians and policy maker for prudent use of antimicrobials in dogs and cats.

Comment:

- According to Table 1, the percentage of samples with S. pseudintermedius isolated in dogs and caste is quite similar (ranged 31-36%), please discuss what are the possible explanation why the prevalence of MRSP in cats is significantly higher than found in dogs.

- Figure 3: Please add the cut-off value that was used of each animicrobial in the figure legend or inside the figure.

Author Response

Comment 1: According to Table 1, the percentage of samples with S. pseudintermedius isolated in dogs and caste is quite similar (ranged 31-36%), please discuss what are the possible explanation why the prevalence of MRSP in cats is significantly higher than found in dogs.

Response 1: Thank you for your comment. We added this part in the discussion: “Notable is the large difference in MRSP prevalence between dogs and cats. S. pseudintermedius seems to be uniquely adapted to dogs, which is a common member of the canine flora and a common pathogen for skin, urinary tract and other infections in dogs [9]. We can only speculate that this adaption changes the pathomechanism of S. pseudintermedius infections in dogs, resulting indirectly in lower resistance rates. This difference however does not seem to apply to wound samples (see 3.2 below). Especially for pyoderma one can assume a difference in pathomechanism between dogs and cats, with canine pyoderma being one of the most common reasons for presentation in a veterinary practice [9]. Other factors, including differences in the antibiotics used for each species and the tendency of cats to visit the vet only when they are seriously ill, as they often hide their symptoms, can also have an effect on the resistance rate. As there are hardly any studies comparing the MRSP rates of cats and dogs, we can only discuss possible influencing factors to a limited extent. Further research is needed to determine whether MRSP prevalence is consistently higher in cats and to explore the underlying reasons.” (Page 6, Discussion, 3.1. MRSP prevalence in cats and dogs, Line 208 ff.)

For that we furthermore added values in the results part:

S. pseudintermedius was isolated in the following proportions across different sample types: 33.2% from skin/soft tissue samples (Dogs: 41.3%, Cats: 4.6%), 22.2% from wound samples (Dogs: 32.5%, Cats: 6.3%), 15.3% from respiratory tract samples (Dogs: 29.8%, Cats: 2.4%), 15.3% from urogenital tract samples (Dogs: 44.8%, Cats: 1.9%), and 25.3% from other samples (Dogs: 27.6%, Cats: 3.5%). Detailed sample counts for S.pseudintermedius isolation categorized by specific organ systems are presented in Table 1. (Page 3, Results, Line 103 ff.)

Comment 2: Figure 3: Please add the cut-off value that was used of each animicrobial in the figure legend or inside the figure.

Response 2: Thank you for addressing this. We added the cut-off values in the figure legend.

Figure 3. Resistance to various antimicrobial substances within all canine and feline S. pseudintermedius samples (n=44,880) compared between species. MIC interpretation (S – I – R) was performed following the CLSI guidelines referencing documents Vet01S-Ed6 and M100Ed33 [20,21]. The breakpoints used were: Ampicillin (Canine and feline urinary tract infections (UTI) and feline non_UTI): S ≤ 0.25, R ≥ 1, (Canine non-UTI): S ≤ 0.25, R ≥ 0.5; Cephalexin (canine and feline UTI and non-UTI): S ≤ 2, R ≥ 4; Chloramphenicol (canine and feline UTI and non-UTI): S ≤ 8, R ≥ 32; Clindamycin (canine and feline UTI and non-UTI): S ≤ 0.5, R ≥ 4; Enrofloxacin (canine and feline UTI and non-UTI): S ≤ 0.5, R ≥ 4; Gentamicin (canine and feline UTI and non-UTI): S ≤ 4, R ≥ 16; Doxycycline (canine and feline UTI and non-UTI): S ≤ 4, R ≥ 16; Rifampicin (canine and feline UTI and non-UTI): S ≤ 1, R ≥ 4; Sulfamethoxazole + Trimethoprim (canine and feline UTI and non-UTI): S ≤ 2/38, R ≥ 4/76; Abbreviations: AST: antimicrobial susceptibility testing, S: susceptible, I: intermediate, R: resistant. 

(Page 5, Results, Figure 3, Line 150 ff.)

Reviewer 2 Report

Comments and Suggestions for Authors

The manuscript is well written and easy to understand, and may be accepted in this present form.

Author Response

Comment 1: The manuscript is well written and easy to understand and may be accepted in this present form.     

Response 1: Thank you very much for taking the time to review our manuscript and for your kind words.

Reviewer 3 Report

Comments and Suggestions for Authors

General evaluation:

Authors evaluated the prevalence and the antimicrobial profiles of methicillin-resistance Staphylococcus intermedius isolated from samples of different origin of dogs and cats from 2019 to 2021. The study highlights different AMR pattern between dogs and cats, as also among the origin samples.

The entire study is well written and structured and add relevant information about the circulation, the trend and prevalence of AMR of MRSP in Germany.

I included some comments below.

-          Line 159: I suggest to change “contrasting” with “monitoring”.

-          Material and methods section: did author use MALDI-TOF-MS to confirm the identification or as the only species identification test? A classical biochemical method identification of strains should be mandatory before any other tests. If only MALDI-TOF-MS was used, it should be included in the limitations of the study.

Author Response

Comment 1:  Line 159: I suggest to change “contrasting” with “monitoring”.

Response 1: Thank you for your suggestion. We chose to emphasize the contrast to highlight the fact that we have data from both species, dogs and cats, allowing us to compare the MRSP prevalence between them. We hope this approach is acceptable to you.

Comment 2: Material and methods section: did author use MALDI-TOF-MS to confirm the identification or as the only species identification test? A classical biochemical method identification of strains should be mandatory before any other tests. If only MALDI-TOF-MS was used, it should be included in the limitations of the study.

Response 2: Thank you for pointing that out. We checked with the laboratory and added the missing identification tests in the methods section.

"Bacterial species were identified by the laboratory based on culture morphology, hemolysis and MALDI-TOF-MS (Bruker Daltonics, Bremen, Germany) following established protocols." (Page 9, Material and Methods, Line 315 ff.)